# New Insight on Antibiotic Resistance and Virulence of *Escherichia coli* from Municipal and Animal Wastewater

**DOI:** 10.3390/antibiotics10091111

**Published:** 2021-09-14

**Authors:** Gabriela Gregova, Vladimir Kmet, Tatiana Szaboova

**Affiliations:** 1The University of Veterinary Medicine and Pharmacy in Košice, Komenského 87, 040 01 Košice, Slovakia; tatiana.szaboova@uvlf.sk; 2Centre of Biosciences, Slovak Academy of Sciences, Institute of Animal Physiology, Šoltésovej 4, 040 01 Košice, Slovakia; kmetv@saske.sk

**Keywords:** *E. coli*, animal wastewater, municipal wastewater, resistance, ESBL

## Abstract

Antibiotic resistance of the indicator microorganism *Escherichia coli* was investigated in isolates from samples collected during the course of one year from two wastewater treatment plants treating municipal and animal wastes in Slovakia, respectively. The genes of antibiotic resistance and virulence factors in selected resistant *E. coli* isolates were described. A high percentage of the isolates from municipal and animal wastewater were resistant to ampicillin, streptomycin, tetracycline, ceftiofur, ceftriaxone, and enrofloxacin. In the selected *E. coli* isolates, we detected the following phenotypes: ESBL (20.4% in animal wastewater; 7.7% in municipal wastewater), multidrug-resistant (17% of animal and 32% of municipal isolates), high resistance to quinolones (25% of animal and 48% of municipal samples), and CTX-M (7.9% of animal and 17.3% of municipal isolates). We confirmed an integro-mediated antibiotic resistance in 13 *E. coli* strains from municipal and animal wastewater samples, of which the *Tn3* gene and virulence genes *cvaC*, *iutA*, *iss*, *ibeA*, *kps*, and *papC* were detected in six isolates. One of the strains of pathogenic *E. coli* from the animal wastewater contained genes *ibeA* with *papC*, *iss*, *kpsII*, *Int1*, *Tn3*, and *Cit.* In addition, one bla_IMP_ gene was found in the municipal wastewater sample. This emphasises the importance of using the appropriate treatment methods to reduce the counts of antibiotic-resistant microorganisms in wastewater effluent.

## 1. Introduction

A wastewater treatment plant (WWTP) presents an important point in the protection and environmental hygiene of the urban and rural environment. Adequate treatment can minimise the discharge of many water contaminants, mostly organic substances, chemicals, and microorganisms [1,2,3].

Many contaminants are not completely eliminated from the wastewater by treatment processes used in WWTPs. The degradation of hormones, pesticides, antibiotics, antihistaminic, and drugs is only limited, and they are commonly present in the aquatic environment [2,3].

Antibiotic-resistant organisms from wastewater enter the environment and spread resistance among water-indigenous microbes [4]. 

Antimicrobial resistance poses a serious global threat to humans, animals, and the environment. Deficiencies in the treatment of wastes that may contain resistant agents of bacterial infections may result in many human deaths (estimation for 2050 is 10 million human deaths) [5]. 

Households, hospitals, pharmaceutical plants, animal farms, veterinary clinics, and other health care facilities represent the main source of antimicrobial agents in wastewater [1,6].

*E. coli* is a conventional inhabitant of the intestines of most animals, including humans. It is frequently associated with multiple antimicrobial resistance and is the main cause of bacterial intestinal and extra-intestinal infections (diarrhoea, urinary tract infections, septicaemia, and neonatal meningitis) in livestock and humans [7]. 

The main current and future risks to the human population arise from resistant *E. coli* with extended spectrum beta-lactamases (ESBLs), including cefotaximases (CTX-M), AmpC beta-lactamases, carbapenemases (KPC), and plasmid quinolones (PMQR). ESBL producing *E. coli* is now relatively common and often exhibits multidrug resistance. The increase in ESBL producing *E. coli* strains in recent years has been observed in human infections and among bacteria isolated from food animals, such as cattle, pigs, and poultry [8,9,10].

Bacterial inactivation procedures in WWTPs are unable to deactivate intracellular genes of resistance [11,12], which can persist even after chlorination [13], although the sequential use of chlorination and UV irradiation may improve the antibiotic resistance genes’ (ARG) inactivation [14,15].

These dangerous resistant bacteria originating from the WWTPs could be discharged into water recipients and be emitted into the air and area surrounding the WWTPs. This means an increased occupational health risk for employees of WWTP and the risk of the transmission of infectious diseases to waterborne organisms [16]. 

The aim of this study was to compare the antibiotic resistance in *E. coli* isolated from animal and municipal wastewaters.

The study provides evidence of the diversity of antimicrobial resistance, genetic lineages, and virulence factors of *E. coli* isolated from wastewater samples. 

## 2. Results

### 2.1. Antimicrobial Susceptibility Profiles

A modified microdilution method with the VetMIC panel was used to detect the antimicrobial resistance in 88 *E. coli* strains isolated from animal wastewater (WW) and 108 strains from municipal wastewater. 

A comparison of the percentage of resistance from animal and municipal wastewater revealed a higher antibiotic resistance of municipal wastewater *E. coli* isolates (Figure 1). Only the resistance to ceftazidime was slightly higher in samples from animal wastewater (2% vs. 1%). 

The highest occurrence of resistance was detected for ampicillin (85% in municipal WW and 65% in animal WW), followed by nalidixic acid (68% vs. 46%), tetracycline (61% vs. 61%), cotrimoxazole (63% vs. 37%), and streptomycin (both 55%). The isolates from the municipal wastewater contained a higher percentage of resistance to florfenicol (36.5% compared to 16% in animal wastewater), chloramphenicol (32% vs. 11.4%), and cotrimoxazole (63.5% vs. 37.5%). The percentage of resistance to ciprofloxacin was 48% for municipal WW and 26% for animal WW. The percentage of resistance to enrofloxacin was almost 50% in municipal WW and 32% in animal WW. Almost all strains were susceptible to ertapenem (3% resistance of municipal isolates, MICxG 0.1 mg/L) and colistin.

Figure 2 shows the results of MICxG of some betalactams and fluoroquinolones from the animal and municipal wastewater. Compared to the isolates from the animal and municipal wastewater, a higher level of MICxG was detected in isolates from the municipal wastewater (Figure 2). MICxG of ceftriaxone was 5.5 mg/L for *E. coli* from the municipal WWTP but only 2.4 mg/L for *E. coli* from the animal WWTP. 

Of the isolates from the animal and municipal wastewater, MICxG 1.1 mg/L and 2.8 mg/L, respectively, were detected for ceftiofur. MICxG of cefquinome was detected at the level of 2.2 mg/L in municipal WW and 1.7 mg/L in animal WW, while ceftazidime was detected at 0.9 mg/L vs. 0.7 mg/L. Furthermore, in isolates from the municipal wastewater, a higher MICxG of fluoroquinolones was detected.

Selected *E. coli* isolates exhibited diverse resistance phenotypes in the animal and municipal wastewater (Figure 3). The results express higher resistance in *E. coli* isolated from the municipal wastewater. Of the *E. coli* isolates recovered from the animal and municipal wastewater, 17% and 32.6%, respectively, were identified as multidrug-resistant phenotypes (selected for the PCR analysis). Furthermore, 25% of animal and 48% of municipal *E. coli* isolates from the wastewater were defined as having a high level of the quinolone phenotype. Respectively, 7.9% and 20.4% of the *E. coli* isolates from the animal wastewater were defined as from the CTX-M group and ESBL TEM phenotype, respectively. The municipal wastewater exhibited a lower percentage of ESBL TEM phenotypes (7.7%) but a higher percentage of multi-resistant quinolones (48%) and CTX-M group phenotypes (17.3%). These results indicate the high risk posed by both WWTPs. Thus, it is very important to use appropriate treatment methods capable of reducing the numbers of antibiotic-resistant microorganisms in the effluent and limiting the risk to the environment.

### 2.2. Genotyping Resistance, Mobilon, and Virulence Factors Detection

The study provides evidence on the diversity of antimicrobial resistance, genetic lineages, and virulence factors of *E. coli* isolated from the animal and municipal wastewater treatment plants. The *E. coli* strains with the most interesting combination of properties (multidrug-resistant phenotype, ESBL TEM, and CTX-M phenotype) are presented in Table 1.

The PCR analysis revealed the presence of CTX-M1 group genes and CIT genes that were associated with a class 1 integron.

In the animal WW isolates, plasmid-mediated quinolone resistance (*qnr*S), eflux genes (*oqxA*, *oqxB)*, and genes of resistance (*qepA*, *gnrA*, *qnrB*, *qnrS*, *aac(6**′**)IbCr)* were not detected.

There was a clear correlation between *chuA* positivity (pathogen group B2), *iutA*, and *kpsII* presence in the human urinal strains. However, only one third of the environmental strains belonged to the pathogen groups B2 and D (Table 1).

We confirmed integron-mediated antibiotic resistance in the selected isolates. Integron cassette class 1 was confirmed in 13 *E. coli* strains. Of those, eight transposon Tn3 genes, as well as six virulence genes, namely *cvaC*, *iutA*, *iss*, *ibeA*, *kps*, and *papC*, were detected in six isolates. The genes *cvaC*, *iutA*, *iss*, and *papC* were the most frequently detected virulence genes in the *E. coli* strains. One of the pathogenic *E. coli* (B2) strains isolated from the animal wastewater contained the genes *ibeA* with *papC*, *iss*, *kpsII*, integron 1, and transposon 3, as well as CIT group genes.

Genes of CTX-M1, *Cit*, integrase (*Int1)*, transposons (*Tn3)*, *cvaC*, and *iutA* were detected in two strains of *E. coli* from the municipal wastewater samples.

The IMP gene was detected in one *E. coli* isolate from the municipal wastewater sample.

## 3. Discussion

The comparison of the antimicrobial resistance of *E. coli* isolates from the animal and municipal wastewater samples showed differences in the percentages of the resistance and occurrence of virulence genes, mobile elements, and phylogenetic groups of *E. coli* strains.

The treatment processes in the WWTPs can partly remove antimicrobials from the wastewater but, at the same time, they can increase the pressure on the bacterial community and the selection of multi-resistant bacterial species [17].

Schroeder et al. [18] tested the resistance of *E. coli* isolates of human and animal origin and found that approximately half of the isolates displayed resistance to one or more antimicrobials, including penicillins, sulphonamides, cephalosporins, tetracyclines, and aminoglycosides, with the highest frequencies of antimicrobial resistance in humans and turkeys, and the lowest in non-food animals.

The use of antimicrobials can support health and high productivity in intensive animal production, but can also contribute to the emergence and spread of antimicrobial resistance [19]

According to the HP-CIA classification, antimicrobials of category B are defined as the compounds with the highest-priority. Critically important antibiotics are used in veterinary medicine only as a last resort after sensitivity testing has been conducted and when no other antibiotic was found to be clinically effective. Category B antimicrobials includes the third generation of cephalosporins (ceftiofur and cefoperazone), fourth generation of cephalosporins (cefquinome), fluoroquinolones (enrofloxacin), and polymyxins (colistin). The administration of these antimicrobials in veterinary medicine may contribute to the development, selection, and proliferation of resistance to clinically relevant antimicrobials because of cross-resistance and horizontal gene transfer [20].

The use of antimicrobials in veterinary practice in Slovakia decreased in recent years in contrast with their use in human medicine. This reflects antibiotic resistance and the transfer of resistant genes in both wastewater and then into the environment. In recent years, the antibiotics tetracyclines (19.3 mg/population correction unit PCU), penicillins (9.2 mg/PCU), sulfonamides (5.8 mg/PCU), pleuromutilins (3.8 mg/PCU), and fluoroquinolones (3.0 mg/PCU) were the most used in animal production in Slovakia. The total amount of antibiotics used in Slovakia was 49.3 mg/PCU. The use of antibiotics is slightly under the EU median. Consumption of pleuromutilins and fluoroquinolones is double compared to the EU median. In 2018, the overall sale of ATB in Slovakia, used mainly in food-producing animals, was 12.4 in tonnes of the active ingredient [20].

Treatment for *E. coli* infection has become increasingly complicated by the emergence of resistance to the first-line antimicrobial agents, including fluoroquinolones [6].

Fluoroquinolones (FQs) are used extensively in both human and veterinary medicine, considered as important weapons against Gram-negative and Gram-positive organisms due to their ability to selectively inhibit bacterial DNA synthesis. They are the third largest group of antibiotics, accounting for 17% of the global market [21].

In our study, we detected a high level of fluoroquinolone resistance in both the animal (26% ciprofloxacin and 31% enrofloxacin) and municipal wastewater isolates (45% ciprofloxacin and 50% enrofloxacin).

A high concentration of fluoroquinolones is excreted into the wastewater by urine (45–62%) or faeces (15–25%) without being metabolised [22]. According to Kümmerer [23], fluoroquinolones are not easily biodegradable and the mechanism for removing them from the environment may be associated with adsorption onto sludge during biological treatment processes [24].

Plasmids, integrons, and transposons are mobile genetic elements often involved in horizontal gene transfer processes [25]. They were found in almost all *E. coli* samples from both investigated sites.

The occurrence of multidrug-resistant bacteria or *Enterobacteriaceae* that produce ESBLs carrying pAmpC, blaOXA-1, and blaTEMs into wastewaters appears to be a serious threat. The production of ESBLs in animal wastewater, which are bacteria resistant to fluoroquinolones, is of high concern.

Savin et al. [26] compared the resistance of *E. coli* isolates recovered from the poultry and pig slaughterhouses. They found a combined resistance expressed to piperacillin (PIP) and cefotaxime (CTX) in 51.4%, and a combination of ceftazidime (CAZ) and ciprofloxacin (CIP) in 58.3%. The rates of resistance to combinations of β-lactam and β-lactamase inhibitors were between 2.9% and 17.1%. The highest rate was observed for ceftolozane-tazobactam. The authors also found 17.1% and 19.4% resistance to colistin in the isolates from the poultry and pig slaughterhouses, respectively. Resistance to carbapenems (imipenem and meropenem), amikacin, and tigecycline was not found in the analysed isolates.

According to Clermont et al. [7], *E. coli* strains belong to four main phylogenetic groups (A, B1, B2, and D). The virulent extra-intestinal strains belong mainly to group B2 and, to a lesser extent, to group D, while most commensal strains belong to group A.

Avian intestinal strains of *E. coli* can be transferred directly from birds to humans or could serve as a genetic pool for ExPEC strains. There is a substantial overlap between the phylogroups and virulence factors of *E. coli* from human urinary infections and from *E. coli* associated with avian colibacillosis [27,28].

Sabate et al. [6] analysed and compared the *E. coli* isolated from different wastewater sources. The results provided relevant information about the origin and transmission of extraintestinal pathogenic *E. coli* resistant to antibiotics. The majority of ESBL *E. coli* strains from pigs, cows, chickens, and human wastewater derived from phylogenetic groups A and B1. Chickens were the dominant source of the fluoroquinolone-resistant *E. coli* transmitted to humans.

Selected *E. coli* isolates from poultry slaughterhouses mainly belonged to group B1 (34.3%), followed by F (17.1%), E (14.3%), A and D (each 11.4%), C (8.6%), as well as B2 (2.9%). Most of the isolates from pig slaughterhouses were also assigned to the B1 group (41.7%), followed by A (25.0%), C (22.2%), B2 (5.6%), and D (5.6%) [28].

We confirmed an integron-mediated antibiotic resistance in 13 *E. coli*. Of those, eight isolates carried the transposon *Tn3* gene and virulence genes *cvaC*, *iutA*, *iss*, *ibeA*, *kps*, and *papC* were detected in six isolates. The genes *cvaC*, *iutA*, *iss*, and *papC* were the most frequently detected virulence genes in the *E. coli* strains. One of the pathogenic *E. coli* (B2) strains originating from the animal wastewater contained genes *ibeA* with *papC*, *iss*, *kpsII*, integron 1, and trasposon 3, as well as CIT group genes.

Savin et al. [26] also detected genes coding for virulence factors, *fimH*, and *astA* genes, which were carried in 40% of the *E. coli* isolates from the poultry slaughterhouse and in 16.7% of the isolates from the pig slaughterhouse. A high percentage of isolates carried *iutA*, *iroN*, and *fyuA* genes. One isolate from a pig slaughterhouse carried *hlyD* and *cnf1* genes. The occurrence of *kpsM II* genes was higher in isolates from the poultry slaughterhouses compared to those from the pig slaughterhouses (22.9% vs. 5.6%).

Drugdova et al. [8] revealed that the virulence genes *iutA*, *iss*, *cvaC*, *tsh*, and *papC* were detected significantly more often amongst meat poultry isolates than in faecal strains.

Adegoke et al. [29] analysed *E.*
*coli* isolates from municipal wastewater from South African WWTP and found in influent a large number of pathogenic cefotaxime-resistant *E. coli* that carried ESBL (blaCTX-M and blaTEM) and carbapenem resistance genes (blaKPC-2, blaOXA-1, and blaNDM-1) individually and concurrently. It was found out that they originated from hospitals. This highlights the importance of additional treatments of wastewaters by more effective methods.

In our previous studies, we found similar results of resistance in municipal wastewater. A high incidence of beta-lactams resistance was observed. In biofilm from municipal wastewater, genes of resistance including CTX-M, CMY-2 and *qnrS* were detected. In one ertapenem resistant *E. coli* isolate, the IMP gene and integron 1 was recorded [30]. In wastewater samples from the rendering plant, we formerly detected a high level of resistance to fluoroquinolones, betalactams, CTX-M1, CMY-2, Int 1, and Tn 3 genes. In one sample, the presence of a pandemic clone of *E. coli* ST131 with CMY-2 was present [31].

Recent improvements in WWTPs technologies have focused on the use of electrocoagulation, UV light, and ozonisation. Chen et al. [15] confirmed that electrocoagulation would be an effective method for the removal of both intracellular and extracellular ARGs from the WWTP effluent. With the electrocoagulation electrolysis (time of 60 min under a current density of 20.0 mA/cm^2^ at a neutral pH), selected antimicrobial resistant genes in *E. coli* could be reduced by 1.48–2.61 logs. High current density and pH in the range of 3.0–7.0 could support the removal of resistant genes. The main mechanisms for the removal of resistant genes by electrocoagulation are adsorption and enmeshment of the precipitated flocs. The combination of UV disinfection pre-treatment and electrocoagulation (after 30 min of electrolysis) reduced ARGs by 1.62 to 2.83 logs.

## 4. Materials and Methods

Bacterial diversity and antimicrobial resistance of *E. coli* isolates were monitored in wastewater samples collected during the course of one year from two wastewater treatment plants: one treating municipal wastewater and the other from animal production wastewater.

### 4.1. Characteristics of the Places of Sampling

The municipal wastewater treatment plant is located close to the town of Košice, the second largest town in Slovakia (current population of 240,000), in the eastern part of the country. The treatment process in this WWTP includes mechanical and biological (nitrification–denitrification) stages. The waste sludge is subjected to anaerobic stabilisation with the production of biogas. The WWTP is capable of treating up to 86,400 m^3^ of wastewater per day and 60 tons of sludge undergoes anaerobic stabilisation. The maximum flow through the channels is 1000 L/sec. It processes wastewater produced by 215,000 inhabitants (households, relevant facilities, and precipitation water). The load is equal to 175,000 population equivalents.

The second sampling site was the animal wastewater treatment plant, located on the premises of the only rendering plant in Slovakia that processes animal by-products and carcasses from the whole country. WWTP is the last stage of the technology involved in the safe processing of materials of animal origin.

The treatment in both WWTPs includes several steps within which physical, chemical, and biological processes are used to remove nutrients, inert materials, and pathogens. They are can be described as follows: preliminary treatment (screening and grit removal), primary treatment (gravity sedimentation tanks), and secondary treatment (activated sludge with deep aeration), followed by secondary sedimentation. Fat removal by flotation is part of the primary treatment in the second WWTP. The effluent is discharged into the recipient (river), while the waste sludge is treated anaerobically and then disposed of adequately.

### 4.2. Identification of E. coli Strains and Antibiotic Susceptibility Detection

MacConkey agar (Oxoid, Basingstoke, UK) incubated overnight at 37 °C was used for *E. coli* isolation. Identification of bacteria was performed by a Maldi Tof biotyper (Bruker Daltonics, Germany). One bacterial colony from MacConkey agar was placed on the ground steel target and dried for 15 min. The colony was then overlaid with 2 µL of matrix solution (saturated solution of ɑ-cyano-4-hydroxy-cinnamic acid in 50% acetonitrile with 2.5% trifluoroacetic acid) and allowed to dry for 15 min [10].

The minimal inhibitory concentration (MIC) of ampicillin (AMP), ampicillin with sulbactam (A + IB), ceftazidime (CAZ), ceftazidime with clavulanic acid (CAC), ceftriaxon (CTR), ceftiofur (CFF), cefquinome (CFQ), ertapenem (ETP), gentamicin (GEN), streptomycin (STM), nalidixic acid (NAL), ciprofloxacin (CIP), enrofloxacin (ENR), chloramphenicol (CMP), florphenicol (FLO), tetracycline (TET), cotrimoxazole (COT), and colistin (COL) were determined according to CLSI VET01-S2 [32] and EUCAST [33] by the modified VetMic panel and Miditech system (Bratislava, Slovakia) with interpretative MIC readings [34]. MICxG expresses geometric mean MIC values of an antibiotic agent (mg/L) in E. coli isolates.

### 4.3. Detection of Antibiotic Resistance Genes in E. coli Isolates

Altogether, 88 isolates of *E. coli* were analysed for antibiotic susceptibility and for the presence of ESBLs, pAmpC, and for a high level of fluoroquinolone resistance.

Primers and PCR conditions are listed in Table 2.

The mechanism of ESBLs and the pAmpC phenotype to the β-lactams was interpreted according to the MIC levels of the antibiotics CTR, CAZ, and CAC [32]. Phenotype interpretation of betalactamase resistance mechanisms was performed by interpretative readings of MICs (ampicillins, cephalosporins, and ertapenem) by Livermore et al. [35].

Phenotype interpretation of chromosomal quinolone resistance mechanisms was conducted with modification [9]. High-level resistant MIC for CIP (≥4 mg/L) and ENR (≥16 mg/L) represented three mutations in QRDR (*gyrA* and *papC*).

The presence of *CTX-M* gene groups [36]; bla *CMY* [37]; IMP [38]; integron *Int1*; transposon *Tn3* [39,40]; efflux genes *oqxA*, *oqxB*, and *qepA* [41]; and quinolone resistance genes *qnrA*, *qnrB*, *qnrS*, and *aac**(6’)**IbCr* [42] were determined by PCR and additional resistance genes by DNA microarray. DNA sequencing of PCR products (CTX-M1, M15, and CMY-2) was used.

**Table 2 antibiotics-10-01111-t002:** Used primers and PCR conditions for the detection of resistance genes and mobile elements.

Primer	Sequence from 5′ to 3′	Product bp	Anealing Temperature	References
Mobile Elements				
*Int1*	GGGTCAAGGATCTGGATTTCG ACATGCGTGTAAATCATCGTCG	483	62 °C	[39]
*Tn3*	CACGAATGAGGGCCGACAGGA ACCCACTCGTGCACCCAACTG	4000	58 °C	[40]
**Genes of Resistance**				
CIT	TGGCCAGAACTGACAGGCAAA TTTCTCCTGAACGTGGCTGGC	462	61 °C	[37]
IMP	GAAGGCGTTTATGTTCATAC GTATGTTTCAAGAGTGATGC	587	51 °C	[38]
*qnrA*	ATTTCTCACGCCAGGATTTG GATCGGCAAAGGTTAGGTCA	516	53 °C	[42]
*qnrB*	GATCGTGAAAGCCAGAAAGG ACGATGCCTGGTAGTTGTCC	469	53 °C	[42]
*qnrS*	ACGACATTCGTCAACTGCAA TAAATTGGCACCCTGTAGGC	417	53 °C	[42]
*aac* *(6’)* *IbCr*	GATCTCATATCGTCGAGTGGTGG GAACCATGTACACGGCTGGAC	435	58 °C	[42]
*CTX-M1*	AAAAATCACTGCGCCAGTTC AGCTTATTCATCGCCACGTT	415	52 °C	[36]
*CTX-M 2*	CGACGCTACCCCTGCTATT CCAGCGTCAGATTTTTCAGG	552	52 °C	[36]
*CTX-M9*	CAAAGAGAGTGCAACGGATG ATTGGAAAGCGTTCATCACC	205	52 °C	[36]

### 4.4. Detection of Virulence Factors and Phylogenetic Group Assignment

Virulence factors were detected by PCR (primers and PCR conditions are listed in the Table 3): fimbriae P (*papC*) and temperature-sensitive haemagglutinin (*tsh*); factors associated with complement resistance such as colicin V (*cvaC*), capsule polysacharide K1, and K5 (*kps II*), and increased serum survival (*iss*); and receptors for aerobactin (*iutA*) and *ibeA*, a gene encoding an invasive factor in the ESBL positive environmental *E. coli* (animal waste water), as well as a comparison with human uropathogenic *E. coli*.

*E. coli* isolates were assigned to a pathogen group (B2 and D) and commensal group (A and B1) according to the method by Clermont et al. [7].

## 5. Conclusions

The results of this study proved that environmental *E. coli*, which encodes a mobile genetic element with virulence factors, mobile elements, and ESBLs, could be an important source of resistance for the human population.

The prevalence of *E. coli* isolates that transfer the genes of resistance varies in the observed municipal and animal wastewaters. The antibiotic resistance of *E. coli* and the presence of virulence genes in the municipal wastewater were much higher than in the animal wastewater. The conclusions of the study confirm the need for further investigations of the genes of resistance, virulence factors, and horizontal transfer to microorganisms.

Wastewater treatment plants are a significant source of contamination (recipient water, air, and soil) and may pose a health risk for employees and people living in proximity to the facilities.

This indicates the need for improvements of treatment methods. It is necessary to focus on the development of new effective tertiary treatment methods in order to reduce the risk to the environment and to human health.

## Figures and Tables

**Figure 1 antibiotics-10-01111-f001:**
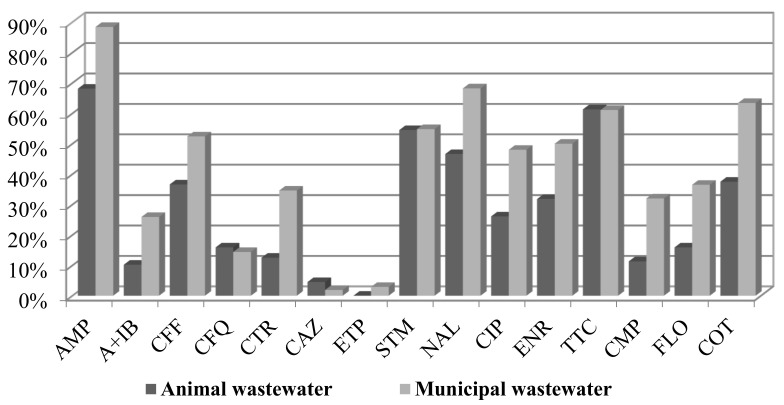
Percentage of resistance of *E. coli* strains isolated from animal and municipal wastewater samples.

**Figure 2 antibiotics-10-01111-f002:**
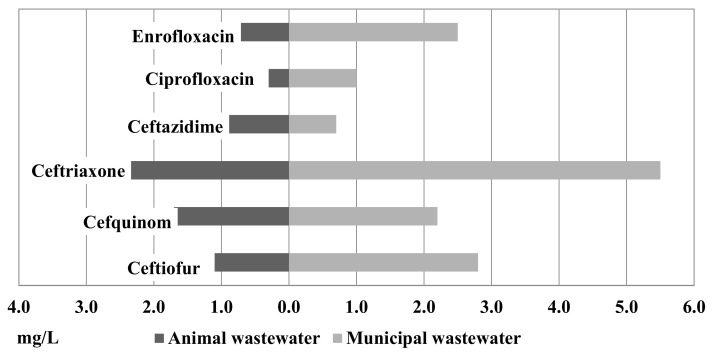
MICxG (mg/L) of betalactams and fluoroquinolones for *E. coli* isolated from the animal and municipal wastewater.

**Figure 3 antibiotics-10-01111-f003:**
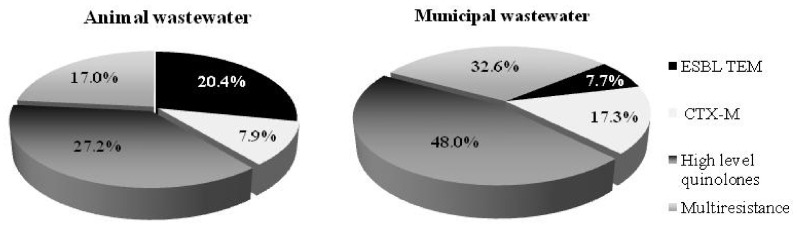
Phenotype resistance of *E. coli* isolates from the animal and municipal wastewater treatment plants.

**Table 1 antibiotics-10-01111-t001:** Resistance genotyping of 14 strains of *E. coli* from the animal wastewater and 13 from the municipal wastewater.

Animal Wastewater (n = 14)			
Commensals (number strains)	PG *	Pathogens (number strains)	PG *
*Cit*, *Int1*, *Tn3* (n = 1)	A	*Cit*, *Int1*, *Tn3* (n = 1)	B2
*Cit*, *Int1* (n = 3)	A	*Cit*, *Int1* (n = 1)	B2
*Int1* (n = 4)	A	*Cit*, *Tn3* (n = 1)	D
*Tn3* (n = 2)	A	*Tn3* (n = 1)	B2
**Municipal Wastewater (n = 13)**			
**Commensals (number strains)**	**PG ***	**Pathogens (number strains)**	**PG ***
CTX-M1, *Int1*, *Tn3* (n = 1)	A	*Int1*, *Tn3* (n = 1)	A
CTX-M1, *Int1*, (n = 2)	A	CTX-M1, *Cit*, *Int1* (n = 1)	B2
*Cit*, *Int1* (n = 2)	A	CTX-M1, *Int1*, *Tn3* (n = 1)	B2
IMP, *Int1* (n = 1)	A	CTX-M1, *Tn3* (n = 1)	B2
qnrS (n = 1)	A	CTX-M1 (n = 1)	D
		CTX-M1, *Cit*, *qnrS*, *Int1*, *Tn3* (n = 1)	nd

PG *–phylogenetic group.

**Table 3 antibiotics-10-01111-t003:** Used primers and PCR conditions for the detection of phylogenetic groups and virulence factors.

Primer	Sequence from 5′ to 3′	Product bp	Anealing Temperature	References
Phylogenetic Group				
*ChuA*	GACGAACCA ACGGTCAGGAT TGCCGCCAGTACC AAAGACA	279	55 °C	[7]
*YjaA*	TGAAGTGTCAGGAGACGCTG ATGGAGAATGCGTTCCTCAAC	211	55 °C	[7]
*TspE4C2*	GAGTAATGTCGGGGCATTCA CGCGCCAACAAAGTATTACG	152	55 °C	[7]
*arpA*	AACGCTATTCGCCAGCTTGC TCTCCCCATACCGTACGCTA	400	59 °C	[43]
*trpA*	AGTTTTATGCCCAGTGCGAG TCTGCGCCGGTCACGCCC	219	59 °C	[43]
**Virulence Factors**				
*iutA*	GGCTGGACATGGGAACTGG CGTCGGGAACGGGTAGAATCG	300	63 °C	[44]
*iss*	GTGGCGAAAACTAGTAAAACAGC CGCCTCGGGGTGGATAA	760	61 °C	[45]
*cvaC*	CACACACAAACGGGAGCTGTT CTTCCCGCAGCATAGTTCCAT	680	63 °C	[44]
*kpsII*	GCGCATTTGCTGATACTGTTG CATCCAGACGATAAGCATGAGCA	272	63 °C	[44]
*tsh*	GGTGGTGCACTGGAGTGG AGTCCAGCGTGATAGTGG	640	55 °C	[46]
*papC*	GACGGCTGTACTGCAGGGTGTGGCG ATATCCTTTCTGCAGGGATGCAATA	328	61 °C	[47]
*oqxA*	GACAGCGTCGCACAGAATG GGAGACGAGGTTGGTATGGA	339	62 °C	[41]
*oqx B*	CGAAGAAAGACCTCCCTACCC CGCCGCCAATGAGATACA	240	62 °C	[41]
*qepA*	CTGCAGGTACTGCGTCATG CGTGTTGCTGGAGTTCTTC	403	60 °C	[48]

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
