# Peer review of "New Insight on Antibiotic Resistance and Virulence of Escherichia coli from Municipal and Animal Wastewater"

_antibiotics, 2021, doi:10.3390/antibiotics10091111_

Round 1
Reviewer 1 Report
This is a revised version of a previously submitted manuscript.
Major concerns:
- The authors do not appear to have a fundamental understanding of the term ESBL and CTX-M or their data are contradictory as there are more CTX-M positive isolates in municipal wastewater yet less ESBLs. This is contradictory observation. I.E., Figure 3.
- MICxG is not defined or why it was used. Figures 1 and 2 also display discrepant data. A certain percentage of isolates were deemed resistant to cephalosporins, yet according to Figure 2, they are mostly susceptible and only a select number of isolates were shown. Not sure why this comparison is made in this manner.
Author Response
Dear reviewer thank you for your opponent review. They were very encouraging and brought very interesting ideas.
English was checked again by native English speaking college and changed all grammatical errors.
Changes in the manuscript are highlighted by yellow colour. Please see attached new manuscript.
|
Q. 1 |
The authors do not appear to have a fundamental understanding of the term ESBL and CTX-M or their data are contradictory as there are more CTX-M positive isolates in municipal wastewater yet less ESBLs. This is contradictory observation. I.E., Figure 3. |
|
Dear reviewer your comment is very good, you are right there was mistake in the Figure no 3. It should be ESBL TEM not only ESBL. Now it is repaired (see line). The Miditech program specifies ESBLs TEM, ESBLs SHV and ESBLs CTX-M for interpreted MIC readings. |
|
|
Q.2 |
MICxG is not defined or why it was used. Figures 1 and 2 also display discrepant data. A certain percentage of isolates were deemed resistant to cephalosporins, yet according to Figure 2, they are mostly susceptible and only a select number of isolates were shown. Not sure why this comparison is made in this manner. |
|
MIC xG we chosen because it express geometric mean MIC values of an antibiotic agent; (mg/L) in E. coli isolates. It is calculated automatically in software Miditech. https://www.mathsisfun.com/numbers/geometric-mean.html |

Reviewer 2 Report
Dear authors
Greetings
This is an interesting manuscript and I would encourage a resubmission of this manuscript and ask you to look at the advices (attached doc reading with adobe). I know this will be disappointing but I think you can produce a better manuscript. Regards

Author Response
Dear reviewer thank you for your opponent review. They were very encouraging and brought very interesting ideas.
English was checked again by native English speaking college and changed all grammatical errors.
Changes in the manuscript are highlighted by yellow colour. Please see attached manuscript.
Reviewer 2
|
Q. 1 |
Line 1 - title 15 words |
|
We are not sure if it will be better shorter title. It express the meaning of the article. |
|
|
Q.2 |
Line 11-12 Where? |
|
Antibiotic resistance of the indicator microorganism Escherichia coli was investigated in isolates from samples collected during one year from two wastewater treatment plants treating municipal and animal wastes in Slovakia, respectively. The place of sampling is written in the part materials and methods |
|
|
Q.3 |
Line 28 – add key words |
|
According your recommendation we add the key words: animal wastewater; municipal wastewater |
|
|
Q.4-6 |
The authors can provide a better graphic for the results |
|
According your recommendations, we changed Figures. We hope it is now better. |
|
|
Q.7 |
Lien 163-165 Add references? |
|
We add reference number (19), and we add it to the list of references. |
|
|
Q.8 |
Line 255-258 The authors can present a better comparison following the previous results |
|
According your recommendation, we improved this paragraph (See line 254-261) |
|
|
Q.9 |
Line 346 Considering the antibiotic-resistance distribution the authors can provide by gene sequences the construction of a phylogenetic tree and outside the represention of genomes I suggest to apply the bioinformatics tools |
|
We are sorry, but we have not data for such analysis. |
|
|
Q.10 |
Line 353-356 the principal differences are ...???? I think the authors can provide better conclusions than this! The authors would can list local risks and health problems reported with human and veterinary importance. |
|
According your recommendation, we improved conclusion (See line 351-365) |

Reviewer 3 Report
It is an interesting short article about antibiotic resistance and virulence of Escherichia coli from municipal and animal wastewater.
Topic is actual in these days.
Article is written correctly, but I have some minor comments:
- number of references in text - please, write without free spaces
- l 53-54 first write full name, than abbreviation in breckets
- full stop after Figure x., and after titles of Figures and Tables
Author Response
Dear reviewer thank you for your opponent review. They were very encouraging and brought very interesting ideas.
English was checked again by native English speaking college and changed all grammatical errors.
Changes in the manuscript are highlighted by yellow colour. Please see attached manuscript.
|
Q. 1 |
Number of references in text - please, write without free spaces |
|
According your recommendation, we changed the writing of references in the text. |
|
|
Q.2 |
Line 53-54 first write full name, than abbreviation in brackets |
|
ESBLs (extended spectrum beta-lactamases) we changed: extended spectrum beta-lactamases (ESBLs). |
|
|
Q.3 |
full stop after Figure x., and after titles of Figures and Tables |
|
We changed it according your notes. |

Round 2
Reviewer 1 Report
The authors have addressed my comments from the previous review cycle.
Author Response
Dear reviewer thank you for your opponent review. They were very encouraging and brought very interesting ideas.
Changes in the manuscript are highlighted by yellow colour. Please see attached manuscript.
Good luck.
Reviewer 2 Report
Dear Authors
Greetings
I appreciate the new version of your work. Amazing! however I insist to suggest that you can improve your graphics (they look scholar) they must to look with a scientific design. Good luck!
Regards
Author Response
Dear reviewer thank you for your opponent review. They were very encouraging and brought very interesting ideas.
Changes in the manuscript are highlighted by yellow colour. Please see attached manuscript.
Good luck.
|
Q. 1 |
I insist to suggest that you can improve your graphics (they look scholar) they must to look with a scientific design. |
|
According your recommendations, we changed Figures. We hope it is better now. |
This manuscript is a resubmission of an earlier submission. The following is a list of the peer review reports and author responses from that submission.
Round 1
Reviewer 1 Report
In this manuscript, the authors characterize E. coli isolates found in municipal and animal wastewater. They attempt to compare susceptibility patterns and compare resistance and virulence genes. Reporting the epidemiological patterns in wastewater is important, but this manuscript fell short.
Major comments:
- Lines 77-80, the authors state that 88 E. coli from animal WW and 108 E. coli from municipal WW were collected, but than only given an MIC for one animal and one municipal isolate. It is not clear, is this the MIC90?
- Again, lines 85-93, are the authors comparing endpoint MICs or MIC90 values or % susceptible vs resistant? Figure 1 shows % resistance to different antibiotics, but it not clear what is present in the text in lines 97-99 and Figure 2. This is not a conventional approach for presenting susceptibility data and is confusing.
- The authors go on to discuss CTX-M and ESBLs. CTX-M is an ESBL. It is not clear what they are differentiating here. Also line 104, high level of quinolones? Do the authors mean quinolone resistance genes? Figure 3 is further perplexing in its presentation.
- Why are only a subset of strains included in Table 1? It is not clear how the commensals are differentiated from the pathogens in this Table.
Minor comments:
- The manuscript needs to be edited for English and grammar throughout.
- What is meant by E. coli being an indicator organism? Do the authors mean, more prevalent?
- beta-lactamase genes should designated as follows "blaIMP"
- Which VetMIC panel was used? GN-mo?
Reviewer 2 Report
The authors attempted to conduct a research that is relevant in the current context when it is important to identify the burden of antimicrobial resistance in our environment. However, there are several critical issues in the manuscript that need to be addressed. Mainly the issues are as follows:
I can see a resemblance between this study and two previous published work from the authors (https://doi.org/10.5604/12321966.1167710 and https://doi.org/10.1038/s41598-020-72851-5) and wonder why no reference to these works were made.
It is not clear to me after reading the introduction section why the study was done. The rationale is not clear. It is advised to rewrite the introduction section and pay special attention on providing evidence to statements and extensively editing this section to rectify the grammatical issues.
Similarly, the method section and result section need extensive work to make them concise and clear. For example, in the result section the authors mention that “A modified microdilution method with the VetMIC panel was used to detected the antimicrobial resistance..” (line 77-78) without mentioning it in the method section. The figures have been reference incorrectly.
A significant work need to be done to make the paper easy to follow through.